

# Vertically-resolved Characteristics of Air Pollution during Two Severe Winter Haze Episodes in Urban Beijing, China

Qingqing Wang[1], Yele Sun[1,2,3*], Weiqi Xu[1,3], Wei Du[1,3], Libo Zhou[1], Guiqian Tang[1], Chen Chen[1], Xueling Cheng[1], Xiujuan Zhao[4], Dongsheng Ji[1], Tingting Han[1,3], Zhe Wang[1], Jie Li[1] & Zifa Wang[1]

[1]State Key Laboratory of Atmospheric Boundary Layer Physics and Atmospheric Chemistry, Institute of Atmospheric Physics, Chinese Academy of Sciences, Beijing, 100029, China
[2]Center for Excellence in Regional Atmospheric Environment, Institute of Urban Environment, Chinese Academy of Sciences, Xiamen, 361021, China
[3]University of Chinese Academy of Sciences, Beijing, 100049, China
[4]Institute of Urban Meteorology, China Meteorological Administration, Beijing, 100089, China

*Correspondence to*: Yele Sun (sunyele@mail.iap.ac.cn)

**Abstract.** We conducted the first real-time continuous vertical measurements of particle extinction ($b_{ext}$), gaseous $NO_2$, and black carbon (BC) from ground level to 260 m during two severe winter haze episodes at an urban site in Beijing, China. Our results illustrated four distinct types of vertical profiles: 1) uniform vertical distributions (37% of the time) with vertical differences less than 5%; 2) higher values at lower altitudes (29%); 3) higher values at higher altitudes (16%), and 4) significant decreases at the heights of ~100 – 150 m (14%). Further analysis demonstrated that vertical convection as indicated by mixing layer height, temperature inversion, and local emissions are three major factors affecting the changes in vertical profiles. Particularly, the formation of Type 4 was strongly associated with the stratified layer that was formed due to the interactions of different air masses and temperature inversions. Aerosol composition was substantially different below and above the transition heights with ~20 – 30% higher contributions of local sources (e.g., biomass burning and cooking) at lower altitudes. A more detailed evolution of vertical profiles and their relationship with the changes in source emissions, mixing layer height, and aerosol chemistry was illustrated by a case study. BC showed overall similar vertical profiles as those of $b_{ext}$ ($R^2$ = 0.92 and 0.69 in November and January, respectively). While $NO_2$ was correlated with $b_{ext}$ for most of the time, the vertical profiles of $b_{ext}/NO_2$ varied differently for different profiles, indicating the impact of chemical transformation on vertical profiles. Our results also showed that more comprehensive vertical measurements (e.g., more aerosol and gaseous species) at higher altitudes in the megacities are needed for a better understanding of the formation mechanisms and evolution of severe haze episodes in China.

## 1 Introduction

Air pollution is a severe environmental problem in China (Chan and Yao, 2008;Zhang et al., 2015), which offers a great challenge for future air quality improvement and economic development. Severe haze episodes with surprisingly high concentration of $PM_{2.5}$ (particles with aerodynamic diameters less than 2.5 μm) dominantly occur in fall and winter seasons.



However, current air quality models often fail to accurately predict extreme haze episodes (Wang et al., 2014b;Zheng et al., 2015a;Wang et al., 2014a;Wang et al., 2014c). One reason is the incomplete understanding of the formation mechanisms of haze pollution. For example, the Weather Research and Forecasting - Community Multi-scale Air Quality (WRF-CMAQ) model showed significant improvements in simulating sulfate and nitrate concentrations and temporal variations in January

2013 episode by incorporating heterogeneous reaction mechanisms (Zheng et al., 2015a), yet it failed to predict the haze peak on January 12–13 that was found to be mainly caused by regional transport (Ji et al., 2014;Sun et al., 2014). However, there are strong arguments on the role of regional transport in the haze pollution (Guo et al., 2014;Li et al., 2015b), especially during severe haze episodes with stagnant meteorological conditions and shallow boundary layer (Sun et al., 2014;Zheng et al., 2015b;Quan et al., 2013). Therefore, vertical characterization of air pollutants is critical for elucidating

the formation and transport of regional haze events, especially for severe haze episodes with boundary layer height less than 300 m (Quan et al., 2013;Tang et al., 2015).

Extensive measurements have been conducted at ground sites for characterization of the composition, sources and formation mechanisms of severe haze episodes in north China (Huang et al., 2014;Ji et al., 2014;Zhao et al., 2013;Sun et al., 2013d). However, vertical measurements are rather limited, particularly in the megacities in north China. Zhang et al. (Zhang et al.,

2009) analyzed the vertical distributions of aerosol number and volume concentrations during 17 aircraft measurements in the spring of 2005 and 2006. Three different types of vertical profiles were observed, which were mainly affected by meteorological conditions. Chen et al. (Chen et al., 2009) further analyzed the vertical measurements of gaseous pollutants in Beijing in summer 2007 and found a significant impact of mountain-valley breeze on the vertical distributions of pollutants. Recent aircraft measurements for black carbon (BC) showed different vertical profiles between southern and northern air

masses, and an enhanced regional transport between 0.5 and 1 km (Zhao et al., 2015). However, most aircraft measurements in the megacity of Beijing were conducted above ~300–500 m, while the vertical distributions in the lower levels of the boundary layer (e.g., <300 m) are rarely characterized. In addition, how the vertical profiles evolve during an entire cycle of haze episode is poorly understood due to the limited aircraft measurements. Although recent measurements with tethered balloons provide more insights into the vertical characteristics of air pollutants (e.g., BC) (Ran et al., 2016;Li et al., 2015a),

most of them were conducted at rural sites, which might be significantly different from urban areas with more complex pollution sources and land surfaces.

The meteorological tower is a unique platform to study the vertical characteristics of air pollutants in the lower levels of the boundary layer. For example, the vertical profiles of trace gases and aerosol species from 3 to 270 m were comprehensively characterized using a 300 m tower at the Boulder Atmospheric Observatory (BAO) during late winter 2011 (Brown et al.,

2013), which provides many new insights into the sources of chemical species (e.g., local emissions, regional transport, and point sources) and aerosol processing at different heights (Kim et al., 2014;Öztürk et al., 2013b). Because the BAO tower is located at a suburban area with low aerosol mass loadings, the vertical profiles of aerosol and gas species (VandenBoer et al., 2013;Riedel et al., 2013;Öztürk et al., 2013a) could be substantially different from those in megacities in China. The



Beijing 325 Meteorological tower (BMT) is located between the north 3$^{rd}$ and 4$^{th}$ ring road in the city center. Compared with the BAO tower, the land surface, sources emissions and atmospheric processes of aerosol particles at the BMT are far more complex. Early studies on the BMT were mainly focused on trace gases, PM$_{2.5}$ and filter measurements (Meng et al., 2008;Sun et al., 2013a;Sun et al., 2009). The results showed that the mixing ratio of O$_3$ often peaked at ~120 m and was

maintained at higher levels in the residual layer at nighttime (Li et al., 2003), while SO$_2$ was found to have the highest mixing ratio at ~50 m (Meng et al., 2008). Filter measurements on the BMT showed higher ratios of organic carbon to elemental carbon at higher altitudes, indicating the vertical differences in secondary formation (Chan et al., 2005). Tao et al. (2007) further found enhanced gas-particle partitioning of polycyclic aromatic compounds at higher heights likely due to the lower temperature and higher relative humidity. However, most previous vertical measurements are either limited by the

time resolution, e.g., 1–3 days for filter collection, or vertical resolution, e.g., typically 3 or 4 heights. Our understanding of the vertical profiles and evolution of air pollutants, and their interactions with the lower boundary layer during severe haze episodes is far from complete. Until recently, Sun et al. (2015b;2016b) conducted real-time simultaneous measurements of aerosol particle composition at ground level and 260 m using two aerosol mass spectrometers. The results showed largely different characteristics of primary and secondary aerosols between ground level and 260 m, elucidating the different

contributions of local emissions and regional transport to air pollution at different heights. In addition, the interactions between meteorological parameters and vertical differences were also illustrated by several case studies. Because these measurements were conducted at two fixed heights, i.e., ground level and 260 m, the vertical variations in the middle were never known, particularly during winter severe haze episodes with the boundary layer height lower than 300 m (Tang et al., 2015).

In this study, we conducted the first continuous vertical measurements of particle extinction ($b_{ext}$), NO$_2$ and BC between ground level to 260 m in the megacity of Beijing during two severe winter haze episodes along with synchronous measurements of aerosol particle composition at ground level and 260 m with two aerosol mass spectrometers. The types of vertical profiles are identified and their relationships with meteorological parameters and aerosol chemistry are elucidated. A more detailed evolution of vertical profiles and their interactions with boundary layer during the severe haze episode is

further illustrated by a case study. Finally, the vertical profiles and relationships between $b_{ext}$, NO$_2$, and BC are investigated.

## 2 Experimental Methods

The vertically resolved measurements were conducted at the Tower Branch of Institute of Atmospheric Physics, Chinese Academy of Sciences using a container that can travel on the BMT at a constant speed of 8 m min$^{-1}$. The instruments including a Cavity Attenuated Phase Shift Particulate Matter extinction monitor (CAPS-PM$_{ext}$, Aerodyne Research Inc.)

(Kebabian et al., 2007), a CAPS-NO$_2$ monitor (Kebabian et al., 2008;Ge et al., 2013), and a 7-wavelength Aethalometer (model: AE33; Magee Scientific Corp.) (Drinovec et al., 2015) were equipped in the container for real-time measurements of light extinction coefficient ($b_{ext}$, λ = 630 nm) of dry fine particles, gaseous NO$_2$, and BC with a time resolution of 1 s, 1 s and



min, respectively. Because all instruments were powered by uninterruptible power supply, the vertical measurements were performed approximately every 4 h from ground level to 260 m during daytime and every 6 h from ground to 200 m during nighttime for safety, with the rest time being ground measurements. In this study, the vertical experiments were conducted during two severe haze episodes that occurred during November 19–21, 2014, and January 12–16, 2015, and in total, 72
vertical profiles were obtained (Table S1, and Figures S1 and S2).

In addition, non-refractory submicron aerosol (NR-PM$_1$) species including organics, sulfate, nitrate, ammonium and chloride, were measured simultaneously at ground level and 260 m on the tower with an Aerodyne High-Resolution Time-of-Flight Aerosol Mass Spectrometer (HR-ToF-AMS) and an Aerosol Chemical Speciation Monitor (ACSM), respectively. The detailed sampling setup and calibration of the two instruments, and data analysis have been described in Xu et al.
(2015b) and Chen et al. (2015). Positive matrix factorization (PMF) (Paatero and Tapper, 1994) was also performed to the mass spectra of organic aerosols (OA). Five OA factors were resolved at both ground site and 260 m, which include three primary OA (POA) factors, i.e., fossil OA (FOA) predominantly from coal combustion emissions, cooking OA (COA), and biomass burning OA (BBOA), and two secondary OA (SOA) factors, i.e., less oxidized oxygenated OA (LO-OOA) and more oxidized OOA (MO-OOA). The mass spectral profiles of five OA factors are presented in Figure S3.

Meteorological variables (temperature ($T$), relative humidity (RH), wind speed (WS), and wind direction (WD)) were measured at 15 heights on the BMT (8, 15, 32, 47, 63, 80, 102, 120, 140, 160, 180, 200, 240, 280, and 320 m) with four-cup anemometers (model O1OC, Met One Instruments), and standard meteoprobe (model HC2-S3, ROTRONIC). The wind field data, $T$, pressure, $CO_2$ and $H_2O$ were measured by ultrasonic anemometers (Gill Instruments Limited, Lymington, UK), and Li-7500$CO_2$/$H_2O$ gas analyzers (LI-COR, Inc., Nebraska, USA) at 7 heights (8, 15, 47, 80, 140, 200, and 280 m) on the
BMT with a time resolution of 10 Hz(Liu et al., 2015). The turbulent kinetic energy (TKE) of motion ($e$) along $x$, $y$, $z$ directions is calculated as $e_u = (u - \bar{u})^2$, $e_v = (v - \bar{v})^2$, and $e_w = (w - \bar{w})^2$, respectively, of which $\bar{u}$, $\bar{v}$, and $\bar{w}$ are the 20 min averages of $u$, $v$, and $w$, respectively. $E_u$, $E_v$ and $E_w$ are then defined as the 20 min averages of $e_u$, $e_v$ and $e_w$, respectively. Virtual potential temperature (VPT) was calculated with meteorological variables measured at 7 heights.

In addition, a Doppler wind lidar (Windcube 200, Leosphere, Orsay, France) was deployed at the same location to measure
wind profiles from 100 to 5000 m with a spatial resolution of 50 m and a time resolution of 10 min (Sun et al., 2016b), and a single-lens Ceilometer (CL51, Vaisala, Finland) was used to measure the vertical attenuated backscatter coefficient (vertical resolution: 10 m). The mixing layer height (MLH) was then retrieved using the algorithm that has been detailed in Tang et al.(Tang et al., 2016) The NR-PM$_1$ aerosol composition and the meteorological variables at ground level were input into the ISORROPIA-II (Nenes et al., 1998) model to predict liquid water content (LWC) associated with inorganic species.



## 3 Results and discussion

### 3.1 Descriptions of the two severe haze episodes

The formation of the two haze episodes in Figure 1 was both initiated by a change in WD from the north-northwest to south-southwest, and then lasted approximately 3–4 days. However, the variations in NR-PM$_1$ were different between the two

episodes. During the first episode in November (Ep1), the mass concentration of NR-PM$_1$ increased rapidly from less than 50 μg m$^{-3}$ to ~250 μg m$^{-3}$ in a day, and then remained at relatively constant levels until a further steep increase on November 21 due to a high RH and LWC event as indicated in Figures 1 and S4. The variations in NR-PM$_1$ were overall consistent with those of H$_2$O which increased from 3 to 5.6 g m$^{-3}$ during this episode (Figure S5). In contrast, the variations of NR-PM$_1$ during the second episode in January (Ep2) were relatively more stable with the concentrations ranging from 100 to 150 μg

m$^{-3}$ until 12:00 on January 15 when steep increases in NR-PM$_1$ at both ground level and 260 m were observed. The average mass concentrations of NR-PM$_1$ during Ep1 were 171 and 125 μg m$^{-3}$ at ground level and 260 m, respectively, which are 30% and 10% higher than those observed during Ep2, indicating that the PM pollution during Ep1 was more severe than during Ep2. A more detailed analysis showed that the higher PM during Ep1 was mainly caused by the lower mixing layer height (mean: 278 ± 109 m) than that during Ep2 (mean: 498 ± 314 m) (Wang et al., 2010). As shown in Figure S4, the MLH

in January was ubiquitously higher than that in November. For example, the MLH was below 400 m for most of the time during Ep1, while it was often above 600 m in daytime during Ep2. This is consistent with a more frequent occurrence of $T$ inversions during Ep1 than Ep2, for example, $T$ inversions were observed at every night from November 19 to 21 (Figure 1d), while only two short periods were observed on January 15 – 16 (Figure 1i).

The temporal variations of NR-PM$_1$ at ground level and 260 m were different between Ep1 and Ep2. While substantial

vertical differences were observed for most of the time during Ep1, the variations of NR-PM$_1$ were overall similar between ground level and 260 m during Ep2. The average vertical difference of NR-PM$_1$ is 46 μg m$^{-3}$ during Ep1, which is much higher than the 18 μg m$^{-3}$ during Ep2. Figure 1 also shows that the vertical difference in NR-PM$_1$ was tightly related to $T$ inversion, and larger differences typically occurred during periods with clear $T$ inversions (Figure 1d). The average composition of NR-PM$_1$ was overall similar between ground level and 260 m during Ep1 and Ep2, except slightly higher

contributions of organics and nitrate at 260 m. While organics dominated NR-PM$_1$ both at ground level (43%) and 260 m (46−47%), the composition of inorganic aerosols was substantially different between Ep1 and Ep2. Particularly, the episode in January showed much higher contributions of sulfate (21−24%) than Ep1 (15−16%). Higher sulfate during Ep2 was likely due to higher RH that facilitated the aqueous-phase processing of SO$_2$ to form sulfate.

### 3.2 Vertical profiles of pollutants

The vertical profiles of $b_{ext}$, NO$_2$ and BC varied substantially during the formation, evolution and clearing stages of the haze episodes, and strongly depended on the vertical changes in meteorological parameters. In this study, we analyzed each vertical profile by calculating the differences in vertical changes (= (max-min)/2×mean, Figure 2), performing linear fit



analysis on the correlations between extinction and height, and checking the presence of $T$ inversions (Figures 2 and 3). Overall, four distinctly different types of vertical profiles were categorized (Figure 4). Type 1 is well mixed as indicated by the small vertical differences (< 5%) and slopes (-0.3 – 0.3 Mm$^{-1}$ m$^{-1}$). Type 2 shows clear decreases as a function of heights, which is mainly categorized according to the large vertical differences (> 5%). In addition, all vertical profiles for Type 2

present negative slopes, and the slope values depend on the absolute extinction values. The vertical profiles of Type 3 are characterized by increases as a function of height, and all the slopes for this type are positive although the vertical differences can vary substantially. The vertical profiles of Type 4 showed significant decreases within a short height interval. Another major difference between Type 4 and Type 3 is the strong $T$ inversion that occurred at a height of less than 150 m. The four types together account for 96% of the total profiles, which can be considered representative during the two haze

episodes. A more detailed summary of the four types of vertical profiles is presented in Tables S2 – S5.

 (1) Uniform vertical distributions. As shown in Figure 4a, the vertical variations for the three vertical profiles of $b_{ext}$ were small, and the vertical differences were all less than 5%, indicating that aerosol particles were relatively well mixed during these three periods. Such types of vertical profiles (37% of the total profiles) were typically observed during daytime, e.g., 10:00–16:00 when $T$ and TKE are high (Figure S5), and MLH is above 260 m (370–1000 m here). Also, there were no $T$

inversions, and the VPT showed small vertical variations (Figure S6) supporting relatively well mixed lower boundary. NO$_2$ showed remarkably similar vertical profiles to $b_{ext}$, which is indicated by the constant ratios of $b_{ext}$/NO$_2$ across different heights (Figure 4a). The vertical profiles of $T$ and RH during these three measurements were similar. While $T$ showed a gradual decrease as the increase with increasing height, RH was relatively even across different heights. Note that the differences of meteorological variables in absolute values were significant, for example, the temperature difference between

V3 (vertical profile of 3) and V24 (vertical profile of 24) is ~6°C, and the wind speed of V24 (~2.5 m s$^{-1}$) is nearly 70% higher than that of V3 (~1.5 m s$^{-1}$). In addition, the TKE for V3 (~0.2 m$^2$ s$^{-2}$) is less than half of that of V24 (~0.6 m$^2$ s$^{-2}$) across different heights (Figure S5), indicating a more stagnant condition for V3. Such meteorological differences might be one of the major reasons for the different $b_{ext}$ levels, e.g., 454 Mm$^{-1}$ for V24 (vertical profile of 24) and 1482 Mm$^{-1}$ for V3, and also largely different $b_{ext}$/NO$_2$ ratios among different vertical profiles (e.g., 8.8 Mm$^{-1}$ ppb$^{-1}$ for V24 and 16.5 Mm$^{-1}$ ppb$^{-1}$

for V3). We note slightly different aerosol composition between ground level and 260 m (Figure 5a) although the vertical differences in $b_{ext}$ and the total mass concentrations of NR-PM$_1$ were small. For example, V24 showed 2% higher contribution of sulfate and 4% lower organics in NR-PM$_1$ at ground level compared with that at 260 m. While secondary formation influenced by RH/$T$ and solar radiation could be slightly different, the relative importance of local emissions versus regional transport at different heights might have also played a role.

(2) Higher values at lower altitudes. Such vertical profiles were frequently observed during this study (29% of the time), which were typically characterized by smooth decreases as a function of height, yet the vertical differences were larger than 5% (Figure 4b). The decreasing rates of $b_{ext}$ varied mostly between 0.4 – 1.4 Mm$^{-1}$ m$^{-1}$ except the profiles during clean periods (e.g., V17). Most of these profiles (60%) occurred at the time between 10:00 – 11:00 and 16:00 – 18:00 when the



vertical convection was moderately high as indicated by $E_w$ in Figure S7. These results suggest that such vertical changes were mainly caused by the dilution effects associated with vertical convection. Another reason is the stronger local emissions at ground level, e.g., cooking emissions during lunch and dinner times (Sun et al., 2013b;Xu et al., 2015a), yet it is slow to be mixed to a higher height, particularly at night with shallower boundary layer. For example, Sun et al.(Sun et al., 2015b)

found much larger ratios between 260 m and ground level for local primary species than regional secondary aerosols at nighttime, indicating that the intensities of local emissions can affect the vertical profiles substantially. Figure 4b shows similar vertical profiles of $T$ and RH, yet the vertical differences of $b_{ext}$ can vary substantially among different vertical profiles. For example, $b_{ext}$ in V29 decreased from 870 Mm$^{-1}$ at ground level to 720 Mm$^{-1}$ at 260 m, which is a change by ~17%. In comparison, $b_{ext}$ in V34 showed a more dramatic decrease by ~34% from 1776 to 1170 Mm$^{-1}$. Although V29 and

V34 were measured at relatively similar time (11:00 and 13:00, respectively), the MLH was substantially different, which was 1106 and 546 m, respectively. The higher MLH associated with higher TKE and stronger vertical convection led to a much smaller vertical gradient for V29 than V34. The vertical profiles of $NO_2$ were similar to those of $b_{ext}$ for most of the time. However, vertical variations in $b_{ext}/NO_2$ ratios were also observed (Figure 4b). For example, the $b_{ext}/NO_2$ ratio in V34 showed a slight decrease from 21 Mm$^{-1}$ ppb$^{-1}$ at ground level to ~20 Mm$^{-1}$ ppb$^{-1}$ at ~150 m, followed by a sudden decrease to

18 Mm$^{-1}$ ppb$^{-1}$. Such a transition height in vertically-resolved $b_{ext}/NO_2$ was consistent with that of $b_{ext}$ and wind direction (Figure 4b), indicating a change of air masses with different pollution characteristics at approximately 150 m.

(3) Higher values at higher altitudes (16% of the time). Figure 4c shows three typical vertical profiles of $b_{ext}$ that increase substantially at approximately 120–150 m. A common meteorological feature for such vertical profiles is the $T$ inversion (Figure 4c) and a change of wind direction at the same height. The heights of VPT with significant changes were generally

similar to those of $T$ (Figure S6), suggesting a stratification of the lower boundary layer at ~150 m. For example, wind direction showed a clear change from the northeast to south between 150–200 m and a further change to the southwest above 200 m during V33 (Figure 4c). The change of WD was associated with an increase in RH, further indicating different air masses below and above the $T$ inversion. As a result, the increase in $b_{ext}$ above ~150 m was mainly caused by regional transport from the south-southwest. While the vertical profiles of $NO_2$ were all similar to those of $b_{ext}$, similar vertical

transition points with the changes in $b_{ext}/NO_2$ ratios were also observed (Figure 4c). As shown in Figure 5c, aerosol composition was very similar between ground level and 260 m before V33, however, it showed increases in the contributions of organics and chloride at 260 m while it remained at small changes at ground level. These results further indicate a change of air masses at high altitude during V33. Compared with V33, V12 also showed an increase in $b_{ext}$ between 120–200 m, which appears to be in contradiction to the much lower NR-PM$_1$ concentration at 260 m (177 µg m$^{-3}$) than ground level (294

µg m$^{-3}$) (Figure 1e). The vertical profile of RH showed consistently high values (~ 70–80%) below 200 m and then a large decrease above 200 m, leading to the presence of a stable layer below 200 m (Figure 1). TKE was ubiquitously less than 0.3 m$^2$ s$^{-2}$, indicating a stagnant atmosphere during this period. Considering the differences between ground level and 260 m, and





the vertical changes in meteorological variables, we expect a rapid decrease of $b_{ext}$ between 200–260 m although the data was not available. The increase of $b_{ext}$ between 120 and 200 m was likely due to a decrease of WS from 2 to 1 m s$^{-1}$.

(4) Significant decreases at the heights of ~100 – 150 m (14% of time). Such vertical profiles were observed dominantly in November (9 of 10, Table S5) which were typically associated with rapid formation or cleaning of PM pollution and $T$

inversions (Figures 1 and 4d). For example, V11 was conducted during the rapid formation stage of the pollution (20:25–20:46, Figure 1e), which was associated with the formation of the high RH/LWC event (Figures 1 and S4). RH and $H_2O$ were consistently higher at lower heights, while they showed sudden decreases at 100 m from ~70% to ~50% (300 m), and 5.5 to 4 g m$^{-3}$ (Figure S5b), respectively. $T$ also showed a strong inversion between 100–200 m with the difference being approximately 1.3°C (Figure 4d). Figure 4d showed that the vertical changes in RH and $T$, and the formation of the higher

RH event was likely due to the interaction of two different air masses that changed from the northeast to northwest at 100 m. Such vertical profiles clearly indicate very different layers at different heights. This is further supported by the vertical profiles of VPT which showed significant increases at similar heights (Figure S6) and suggested a low stable stratification. As a result, the vertical profile of $b_{ext}$ showed a sudden decrease at 100 m, from ~1800 to ~1200 Mm$^{-1}$ at 200 m. Consistently, the mass concentration of NR-PM$_1$ showed a significant increase by ~50 μg m$^{-3}$ at ground level, while it

remained relatively constant at 260 m. The vertical profile of $NO_2$ also showed a similar decrease at the same transition height (from 94 to 83 ppb) as $b_{ext}$, yet the change was much smaller than $b_{ext}$, leading to a similar decrease in vertical profile of $b_{ext}/NO_2$. These results indicate that the local accumulation effects caused by the high RH event are more significant on particles than gaseous species. The vertical changes also have a significant impact on aerosol composition. For example, the contribution of OA to NR-PM$_1$ was higher at ground level (48–51%) than 260 m (44%), mainly due to the accumulation of

local pollutants, e.g., cooking aerosols during the high RH event (Figure 6). Compared with V11, V14 took place during the clearing stage of the pollution (Figure 1e). A stable layer associated with a clear $T$ inversion in the morning was observed at approximately 150 m. RH was significantly different below and above the layer. While RH was consistently high at ~70% below 150 m (Figure 4d), it showed a rapid decrease from 70% to 40% between 150–300 m, suggesting dryer air masses in higher altitudes. Such vertical profiles in RH and $T$ led to a significant transition in $b_{ext}$ at 150 m. While $b_{ext}$ was relatively

constant (~1350 Mm$^{-1}$) below 150 m, it had a sudden decrease at 150 m and reached the lowest value at 260 m (~770 Mm$^{-1}$), which was nearly twice lower than that at low altitudes. Similarly, $NO_2$ showed a sudden decrease from 78 to 67 ppb at the same height, yet the change was much smaller than $b_{ext}$, leading to a significantly different $b_{ext}/NO_2$ ratio below and above 150 m (~18 and ~13–14 Mm$^{-1}$ ppb$^{-1}$, respectively, Figure 4d). The difference is consistent with that of NR-PM$_1$ between ground level and 260 m, which was 175 and 87 μg m$^{-3}$, respectively. Owing to the different air masses and meteorological

conditions, aerosol composition was substantially different between ground level and 260 m, particularly when higher contributions of sulfate and lower contributions of organics were observed at ground level (Figure 5d).



### 3.3 Case study of evolution of vertical profiles

Figure 7 shows a detailed evolution of vertical profiles of $b_{ext}$ and meteorological conditions from 00:00 on November 20 to 17:00 on November 21. The evolution can be classified into six stages, and the types of vertical profiles evolved routinely from Type 4 at nighttime to Type 2 / Type 1 during daytime, and then Type 3/Type 4 at nighttime again. The first stage ($S1$,

00:00–11:00) was characterized by consistent northerly winds and moderately high RH (60–80%). As a result, aerosol species and chemical composition were relatively stable at both ground level and 260 m during this stage. The vertical profile of $b_{ext}$ showed a clear transition height at approximately 120 m at 3:00 am, indicating the presence of a nocturnal stable boundary layer (Stull, 1988). The transition height started to increase with the increase of temperature and reached approximately 200 m at 11:00. Consequently, the vertical profiles evolved from Type 4 to Type 2. The variation of the stable

layer height was consistent with that of MLH which increased from 190 to 350 m. But note that the MLH was ubiquitously higher by ~50 m than the transition height that was determined from $b_{ext}$, which might indicate that the MLH retrieved from the ceilometer measurement was overestimated during severe haze episodes.

The second stage ($S2$) was characterized by a transition of wind direction from north to south at 12:00, which remained unchanged until 20:00. Due to the increasing $T$ and vertical convection, the vertical differences between ground level and

260 m (e.g., NR-PM$_1$ in Figure 7e) were gradually reduced, and the vertical profiles were relatively even and evolved from Type 2 to Type 1, e.g., V9 between 14:00–15:00. This is also consistent with the relatively high MLH, which was generally above 350 m. These results indicate that aerosol particles were relatively well mixed during this stage. Increases in secondary nitrate and SOA were also observed at both ground level and 260 m, which is likely due to the enhanced photochemical production and/or regional transport from the south. The concentration of NR-PM$_1$ at 260 m showed a

significant decrease from ~150 to ~80 µg m$^{-3}$ during stage 3 ($S3$, 16:00–20:00), while it remained relatively constant at ground level. Such changes were likely caused by a significant increase in wind speed between 200–300 m that diluted the pollutants substantially (Figure 7b). This is further supported by significant increases in TKE ($E_u$, $E_v$, $E_w$) above 200 m (Figure S5). Indeed, we observed a significant decrease in NO$_2$ level from ~70 ppb at 200 m to 30 ppb at 260 m, and also a large decrease in BC from 22 to 14 µg m$^{-3}$ for V10 ($b_{ext}$ was not available). A further increase in vertical difference between

ground level and 260 m was observed during the later stage of $S3$ (18:00–20:00), which was primarily caused by significant increase in primary OA (mainly local cooking OA) from ~25 to 130 µg m$^{-3}$ at the ground site (Figure 6b).

After $S3$, $T$ showed a significant decrease while RH showed a rapid increase from 55% to ~80% below 100 m, and LWC at ground surface reached >50 mg m$^{-3}$. The high RH event was mainly caused by the interaction between the warmer southwestern air mass and the colder northeastern one (Figure 7b). The height of the high RH (> 70%) layer was consistently

below 200 m during $S4$ (20:00–22:30). Therefore, aerosol species showed much smaller variations at 260 m compared with the large increases at ground level (Figure 6b). Not surprising, $b_{ext}$ showed a large vertical gradient (Type 4), for example, $b_{ext}$ for V11 decreased from ~2000 Mm$^{-1}$ at ground level to 1200 Mm$^{-1}$ above 100 m. Such a vertical profile was also consistent with a dramatic decrease in MLH from 220 to 135 m. Note that the most significant increases in organics, sulfate,



and chloride during *S*4 likely indicates the rapid accumulation and/or transformation of coal combustion emissions (Wang et al., 2015). The high RH layer height gradually evolved to ~240 m during stage 5 (*S*5, 22:30–9:00) according to the vertical variations in RH, and correspondingly, the MLH increased from 130 to 260 m. Figures 7b and 7c show consistently low WS and high RH at the early stage of *S*5. Although NR-PM$_1$ at ground level showed a gradual decrease, the contributions of

secondary species to NR-PM$_1$ increased. These results likely indicate an enhanced formation of secondary aerosol species due to aqueous phase processing while the scavenging compensated the increases in the total mass. As a comparison, we didn't observe similar increases in secondary aerosol species at 260 m because of low RH (<70%). The vertical profile of $b_{ext}$ showed a transition point at the height of ~100–130 m, overall consistent with the vertical variations in meteorology. After 1:00, an increase in WS above 100 m was observed, leading to a faster decrease of NR-PM$_1$ at 260 m than ground level. The

height of the high RH layer was also decreased from ~240 to 150 m. Such changes in meteorological conditions again led to a strong vertical gradient of $b_{ext}$. For example, $b_{ext}$ for V14 was ~1400 Mm$^{-1}$ at ground level, while it was nearly a factor of two lower at 260 m (~750 Mm$^{-1}$), consistent with the difference in NR-PM$_1$ concentration (170 vs. 90 μg m$^{-3}$). After 9:00, RH quickly dropped to <20% at both ground level and 260 m. Because of the increasing *T*, MLH rapidly increased from <200 to ~550 m (13:00). Consistently, the vertical differences between ground level and 260 m were largely reduced, and the

vertical profiles of $b_{ext}$ showed relatively even distributions (Type 1). These results indicate that aerosol particles during *S*6 became relatively well mixed.

### 3.4 Vertical co-variations of pollutants

Figure 8 shows the correlations of BC and NO$_2$ with $b_{ext}$ for all vertical profiles during the two haze episodes. The vertical profiles of BC were similar to $b_{ext}$ for most of the time ($R^2$ = 0.92 and 0.69 in November and January, respectively). By

performing a linear regression of BC to three POA factors (FOA, COA, BBOA) and secondary aerosol species (SA = SOA + sulfate + nitrate + ammonium) (Sun et al., 2016a), we found that ~30% of BC was associated with SA during the severe haze episode in November, while it was only ~11% during the one in January. This explains the better correlation between $b_{ext}$ and BC in November than in January. The average single-scattering albedo (SSA) calculated with a constant mass absorption efficiency for BC (7.3 m$^2$ g$^{-1}$) (Han et al., 2015b) using Equation (1) was 0.84 and 0.86 in November and January,

respectively.

$$SSA = (b_{ext} - BC \times 7.3) \div b_{ext} \tag{1}$$

The SSA values were generally consistent with those observed in previous studies (Li et al., 2015c;Lee et al., 2007;Han et al., 2015a;Han et al., 2017), e.g., 0.85 – 0.91 during the fall of 2014 (Han et al., 2015a) and 0.85 (±0.04) in the winter of 2015 (Han et al., 2017). We also observed slightly lower values at lower altitudes, indicating an enhanced BC absorption due

to the influences of local emissions (Sun et al., 2015a;Sun et al., 2010). However, for several specific events, we observed very significantly different correlations between BC and $b_{ext}$. For example, V11 showed strong vertical gradients for both BC and $b_{ext}$ with sudden decreases at approximately 100 m (Figure S8). As indicated in Figure 4d, such changes were mainly



associated with corresponding changes in meteorological parameters, which were characterized by a $T$ inversion, a decrease in RH, and a wind direction change from northeast to northwest. The vertical profile of SSA first showed a gradual decrease from 0.82 at ground level to 0.78 at 100 m, followed by a large increase to 0.86 at 200 m (Figure S8). As shown in Figure 9, aerosol composition was substantially different below (ground level) and above (260 m) the transition height. While organics

showed higher contribution at ground site than at 260 m (48 – 51% vs. 44%), the nitrate contribution was relatively lower (18-20% vs. 23 – 24%). Also, OA at the ground site was dominated by primary OA from cooking and biomass burning emissions (70 – 74%), while it was mainly composed of secondary OA (LO-OOA + MO-OOA, ~66%) at 260 m. These results indicate that aerosol particles below the transition height were largely influenced by local source emissions, while those above the height were more affected by secondary aerosols.

$NO_2$ showed overall similar vertical profiles as $b_{ext}$ (Figure S9) during the severe haze episode in November as indicated by the tight correlations in Figure 10 ($R^2 > 0.75$ for 63% of the time). The slopes of $b_{ext}$ versus $NO_2$ were relatively stable before November 21 by varying from 16 $Mm^{-1}$ $ppb^{-1}$ to 20 $Mm^{-1}$ $ppb^{-1}$. However, we observed large decreases in $b_{ext}/NO_2$ from 26 $Mm^{-1}$ $ppb^{-1}$ to 2-3 $Mm^{-1}$ $ppb^{-1}$ on November 21. This is consistent with the fact that the $NO_2$ concentration slowly decreased from ~ 60 to 20 ppb, while $b_{ext}$ rapidly decreased from ~2000 to 60 $Mm^{-1}$. Compared with November, the vertical

profiles of $NO_2$ showed more differences from those of $b_{ext}$ during the haze episode in January. The similar vertical profiles with $R^2 > 0.75$ accounted for 32% of the total profiles (Figure 10). The average $b_{ext}/NO_2$ ratios also varied more significantly (~10 – 20 $Mm^{-1}$ $ppb^{-1}$ for most of the time) than those in November. Also note that the $b_{ext}/NO_2$ ratios in January appeared to show a clear diurnal pattern with higher values at nighttime, which can be explained by the different primary and secondary sources between daytime and nighttime. The types of vertical profiles of $b_{ext}/NO_2$ are relatively similar to those of $b_{ext}$. As

shown in Figure 4, the vertical profiles of $b_{ext}/NO_2$ showed either uniform distributions (Type 1), decreases/or increases as a function of height (Types 2 and 3), or significant decreases at ~100 m (V11) or 180-200 m (V14) (Type 4).

Figures 8 c and d also show that the correlations between $b_{ext}$ and $NO_2$ were not linear, and the ratio of $b_{ext}/NO_2$ appeared to increase as a function of pollution level. For example, the ratio of $b_{ext}/NO_2$ decreased from ~26 to ~17 $Mm^{-1}$ $ppb^{-1}$ during the high RH period of 00:00–7:00 on November 21 as the mass concentration of $NR-PM_1$ decreased from ~300 to 170 μg m$^{-3}$

(Dan et al., 1999;Dan et al., 2003;Mcmurry et al., 2004;Wayne et al., 1991). One explanation is the enhanced formation of secondary aerosol species during severe haze episodes that were associated with the oxidation of precursors (e.g., $SO_2$ and $NO_2$). This is particularly important for $SO_2$ that can be rapidly oxidized to form sulfate via aqueous-phase and/or fog processing (Sun et al., 2013c;Quan et al., 2015), while the role of $NO_2$ oxidation is generally small due to its much slower aqueous-phase processing rates (Seinfeld and Pandis, 2006), which is also supported by the shorter lifetime of $SO_2$ compared

with that of $NO_2$ in the atmosphere. In addition, the scavenging rates of gaseous species and particles can be substantially different during the high RH event (Wayne et al., 1991;Dan et al., 1999;Dan et al., 2003;Mcmurry et al., 2004). While most aerosol species can be efficiently scavenged (e.g., > 50% during fog events with high RH) (Gilardoni et al., 2014), the



scavenging of $NO_2$ is much slower due to its low solubility. This could be another important factor affecting the changes in $b_{ext}$/$NO_2$ ratios.

## 4 Conclusions and implications

The vertical profiles of particle extinction, gaseous $NO_2$, and BC from ground level to 260 m measured during two severe winter haze episodes in the megacity of Beijing varied very dynamically and interacted closely with boundary layer and meteorological conditions. Four types of vertical profiles (96% of the time) were sorted out to elucidate the vertical evolution characteristics of air pollutants during the two severe haze episodes. The diurnal evolution of vertical profiles of $b_{ext}$ highlighted the presence of a nocturnal stable boundary layer between 100–150 m during severe haze episodes and its impact on the vertical characteristics of air pollution. However, the transition heights determined from the vertical changes in $b_{ext}$ were consistently lower than the MLH retrieved from the ceilometer measurements. As shown in Figure 11, the differences between the two methods can be as large as $100 - 150$ m during the periods with the highest $b_{ext}$ (vertical profiles of $11 - 13$) and $\sim 40 - 100$ m during other periods with nocturnal stable boundary layer. These results indicate that the ceilometer measurements might often overestimate the MLH, particularly at nighttime during severe haze episodes, and the vertically resolved measurements on the BMT can be an essential supplement, particularly for the lidar measurements with a blind zone below $\sim$200–300 m. Traditional air quality forecast models often underestimate severe haze episodes substantially. In addition to the incomplete understanding of the formation mechanisms, the very complex and dynamic vertical variations could also be one of the major reasons. Our results also highlight that more comprehensive vertical measurements (e.g., more aerosol and gaseous species) at a higher altitude in the megacities are urgently needed for a better understanding of the formation mechanisms and evolution of severe haze episodes.

## Acknowledgements

This work was supported by the National Key Project of Basic Research (2014CB447900), the National Natural Science Foundation of China (41575120, 41571130034), the Beijing Natural Science Foundation (8161004), National Postdoctoral Program for Innovative Talents (BX201600157), and the General Financial Grant from the China Postdoctoral Science Foundation (2017M610972).

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

**Figure Captions:**

**Figure 1:** Vertical evolution of (a, f) wind direction, (b, g) wind speed, (c, h) relative humidity, and (d, i) temperature during two
severe haze episodes in November and January. (e) and (j) show the time series of NR-PM$_1$ mass concentrations at ground level and 260 m. The two pie charts in each panel present the average composition of NR-PM$_1$ for each episode at ground level (right) and 260 m (left), respectively, and the numbers on the top of pie charts are the average mass concentrations. The number of vertical profile experiments is also marked in (e, j) as green (up) and gray (down) vertical lines, and the time for each vertical profile is detailed in Table S1.

**Figure 2:** Time series of (a) vertical difference (= max – min) and slope of extinction versus height; (b) extinction and vertical difference (= (max-min)/(2*avg)) for each vertical profile during the haze episode in November. The circles refer to the down profiles while boxes indicate the up profiles.

**Figure 3:** Time series of (a) vertical difference (= max – min) and slope of extinction versus height; (b) extinction and vertical difference (= (max-min)/(2*avg)) for each vertical profile during the haze episode in January. The circles refer to the down profiles
while boxes indicate the up profiles.

**Figure 4:** Four selected types of vertical profiles of $b_{ext}$, $b_{ext}/NO_2$, $T$, RH, and WS. "U" in the legend indicates the "up" experiments, the others are the "down" ones. The vertical profile 9 (V9) for $b_{ext}/NO_2$ is missing because NO$_2$ data was not available. The rightmost panels show four selected vertical profiles of WD, i.e., V3, V25, V33, and V11. For clarity, some vertical profiles were offset by certain values as indicated by "+" or "-" in the figure. Note that all vertical profiles reaching 260 m were measured at
daytime, while those of 200 m were measured at nighttime (Table S1).

**Figure 5:** Average chemical composition of NR-PM$_1$ for selected vertical profiles (a) V24, (b) V29, (c) V33, and (d) V14 at ground level (bottom panel) and 260 m (top panel. The numbers on bar charts are the mass fractions of sulfate and organics. "U" and "D" represent "up" and "down" experiment, respectively.

**Figure 6:** Evolution of non-refractory submicron aerosol species and NR-PM$_1$ composition at (a) 260 m and (b) ground level. POA
(= FOA+COA+BBOA) and SOA (= LO-OOA + MO-OOA) were from positive matrix factorization of OA. RH and WD are also shown for a comparison.

**Figure 7:** Evolution of vertical profiles of $b_{ext}$ and meteorological parameters from 00:00 on November 20 to 17:00 on November 21. Also shown are time series of NR-PM$_1$ mass concentrations at ground level and 260 m, and MLH. Note that the vertical profiles of $b_{ext}$ between 16:00–17:30 were not available due to a malfunction of the CAPS PM$_{ext}$, therefore, the two vertical profiles of NO$_2$
were used as surrogates. The shaded areas in (e) indicate the time periods for the vertical measurements.

**Figure 8:** Correlations between BC and $b_{ext}$, NO$_2$ and $b_{ext}$ for all vertical profiles in (a, b) November and (c, d) January. The data points are color coded by the heights. The solid circles in (a) and (b) are data points for V11.

**Figure 9:** Average chemical composition of NR-PM$_1$ and OA at ground level and 260 m for vertical profile 11 (V11). Right panels are the mass fractions of NR-PM$_1$ and OA species. "U" and "D" represent "up" and "down" experiment, respectively.

**Figure 10.** Time series of correlation coefficients ($R^2$) and slopes between $b_{ext}$ and NO$_2$ for each vertical profile in (a) November and (b) January. The circles refer to the down profiles while other are the up profiles.




**Figure 11: A comparison of mixing layer height (MLH) with that estimated from the transition heights of $b_{\text{ext}}$. The numbers of vertical profiles and the differences are also shown in the plot.**

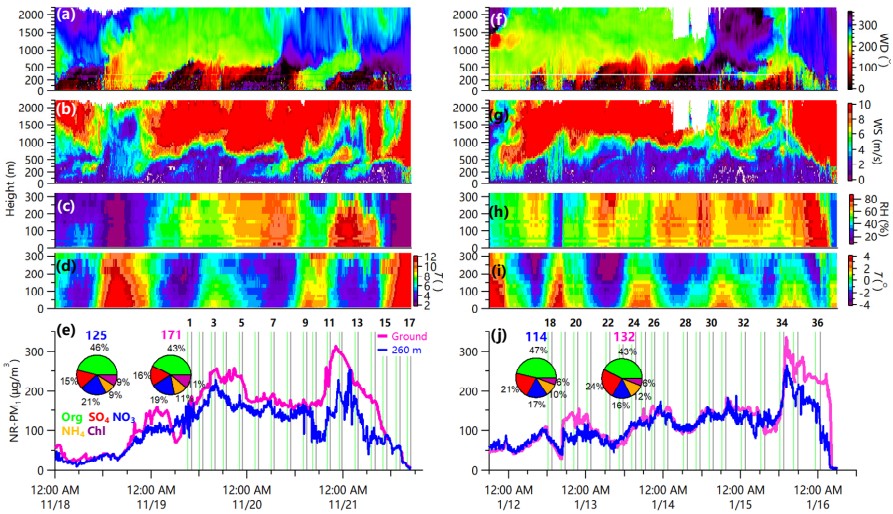

5  **Figure 1: Vertical evolution of (a, f) wind direction, (b, g) wind speed, (c, h) relative humidity, and (d, i) temperature during two severe haze episodes in November and January. (e) and (j) show the time series of NR-PM$_1$ mass concentrations at ground level and 260 m. The two pie charts in each panel present the average composition of NR-PM$_1$ for each episode at ground level (right) and 260 m (left), respectively, and the numbers on the top of pie charts are the average mass concentrations. The number of vertical profile experiments is also marked in (e, j) as green (up) and gray (down) vertical lines, and the time for each vertical**
10  **profile is detailed in Table S1.**





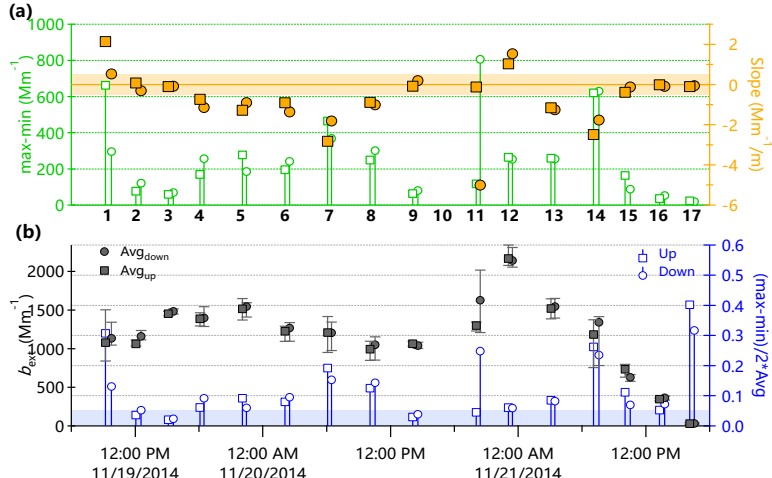

**Figure 2: Time series of (a) vertical difference (= max – min) and slope of extinction versus height; (b) extinction and vertical difference (= (max-min)/(2*avg)) for each vertical profile during the haze episode in November. The circles refer to the down profiles while boxes indicate the up profiles.**

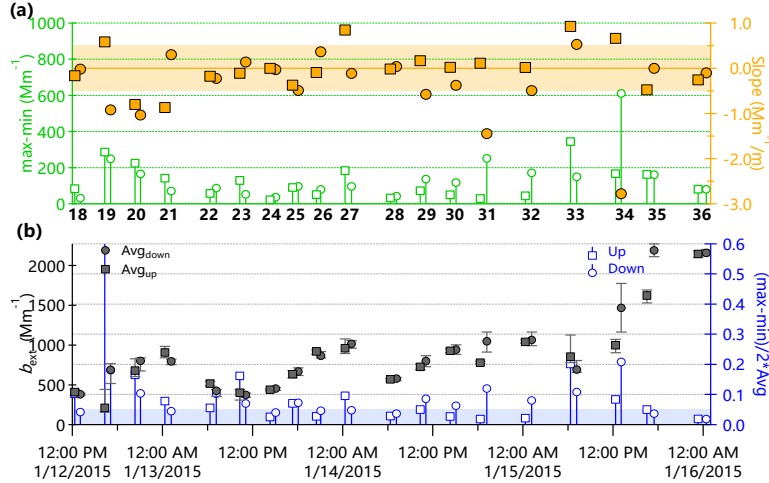

**Figure 3: Time series of (a) vertical difference (= max – min) and slope of extinction versus height; (b) extinction and vertical difference (= (max-min)/(2*avg)) for each vertical profile during the haze episode in January. The circles refer to the down profiles while box indicate the up profiles.**



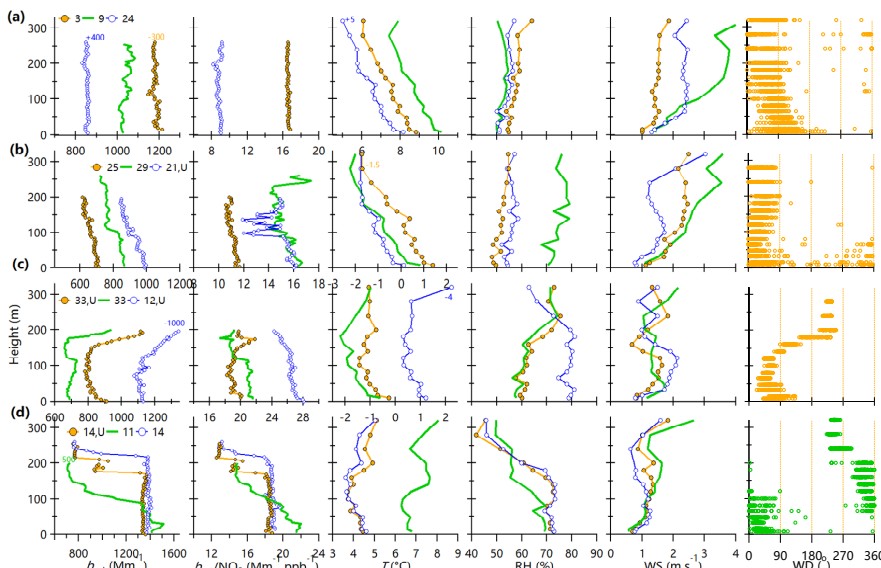

**Figure 4: Four selected four types of vertical profiles of $b_{ext}$, $b_{ext}/NO_2$, $T$, RH, and WS. "U" in the legend indicates the "up" experiments, the others are the "down" ones. The vertical profile 9 (V9) for $b_{ext}/NO_2$ is missing because NO₂ data was not available. The rightmost panels show four selected vertical profiles of WD, i.e., V3, V25, V33, and V11. For clarity, some vertical profiles were offset by certain values as indicated by "+" or "-" in the figure. Note that all vertical profiles reaching 260 m were measured at daytime, while those of 200 m were measured at nighttime (Table S1).**



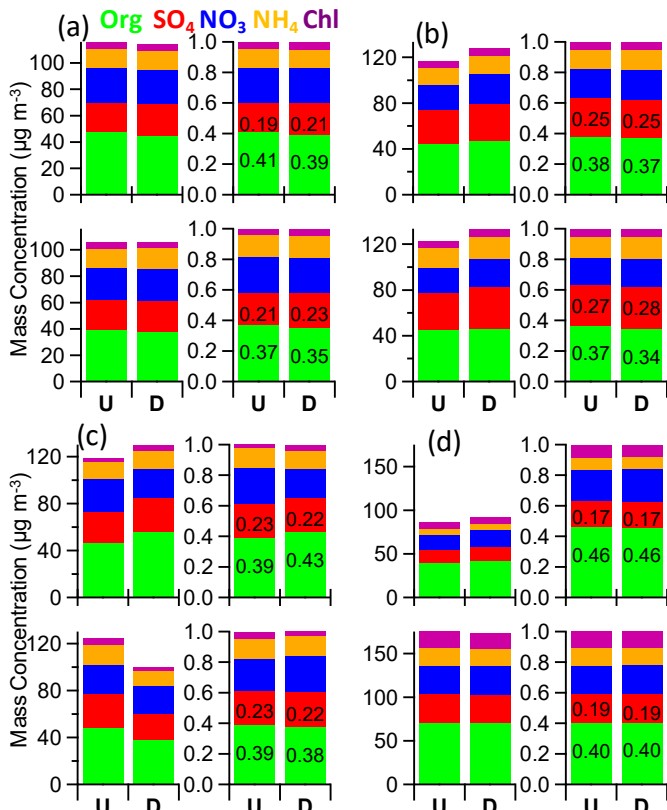

**Figure 5:** Average chemical composition of NR-PM$_1$ for selected vertical profiles (a) V24, (b) V29, (c) V33, and (d) V14 at ground level (bottom panel) and 260 m (top panel. The numbers on bar charts are the mass fractions of sulfate and organics. "U" and "D" represent "up" and "down" experiment, respectively.



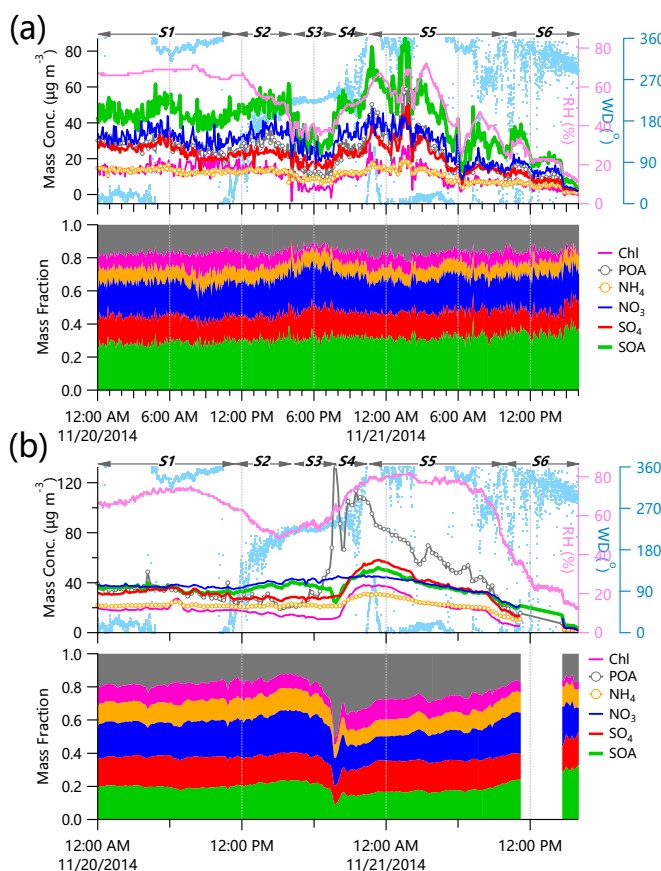

**Figure 6: Evolution of non-refractory submicron aerosol species and NR-PM$_1$ composition at (a) 260 m and (b) ground level. POA (= FOA+COA+BBOA) and SOA (= LO-OOA + MO-OOA) were from positive matrix factorization of OA. RH and WD are also shown for a comparison.**





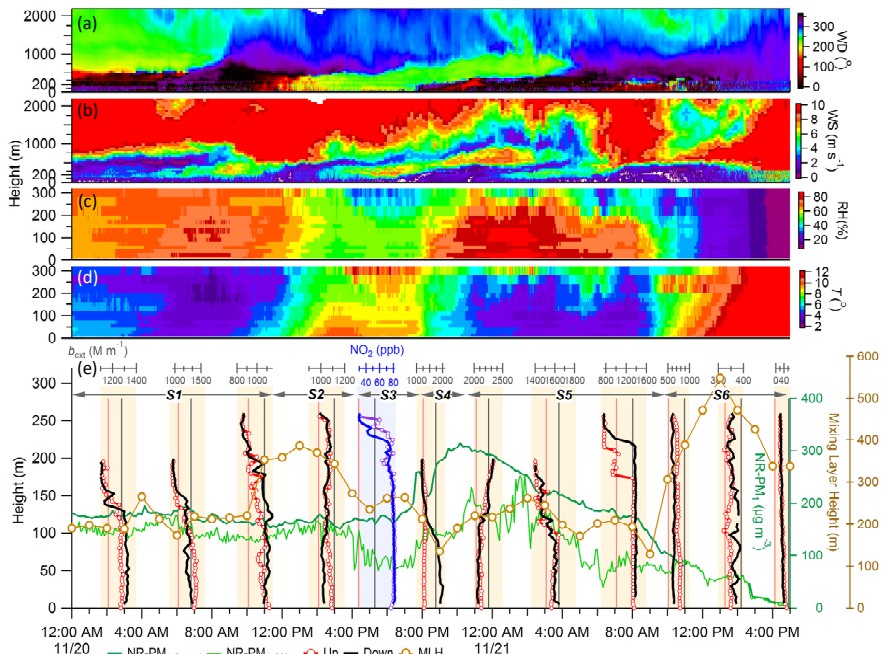

**Figure 7: Evolution of vertical profiles of $b_{ext}$ and meteorological parameters from 00:00 on November 20 to 17:00 on November 21. Also shown are time series of NR-PM$_1$ mass concentrations at ground level and 260 m, and MLH. Note that the vertical profiles of $b_{ext}$ between 16:00–17:30 were not available due to a malfunction of the CAPS PM$_{ext}$, therefore, the two vertical profiles of NO$_2$ were used as surrogates. The shaded areas in (e) indicate the time periods for the vertical measurements.**



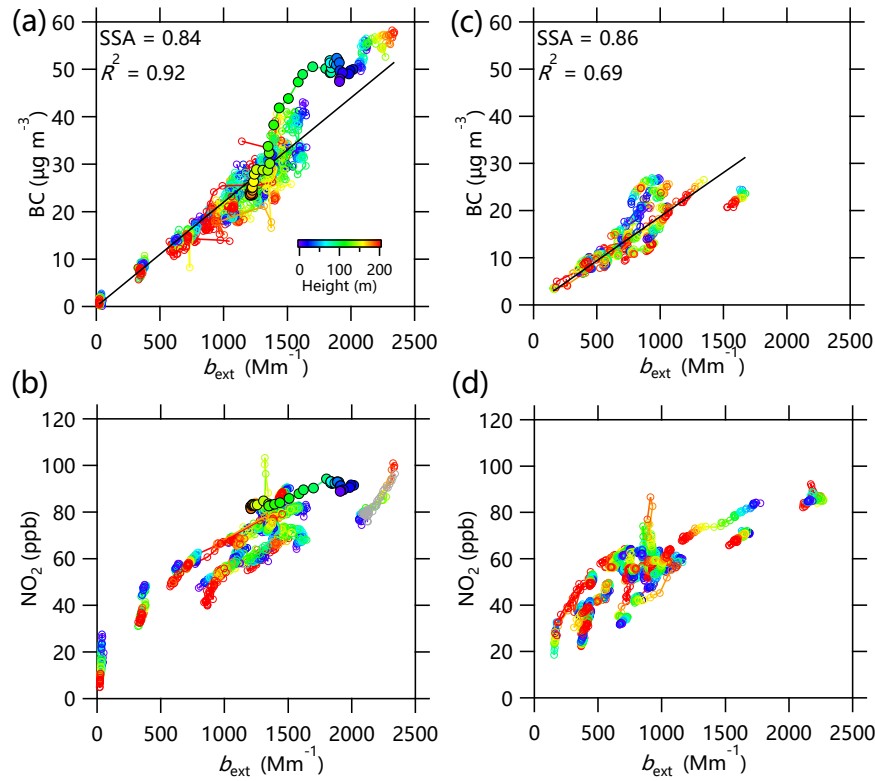

**Figure 8: Correlations between BC and $b_{ext}$, $NO_2$ and $b_{ext}$ for all vertical profiles in (a, b) November and (c, d) January. The data points are color coded by the heights. The solid circles in (a) and (b) are data points for V11.**



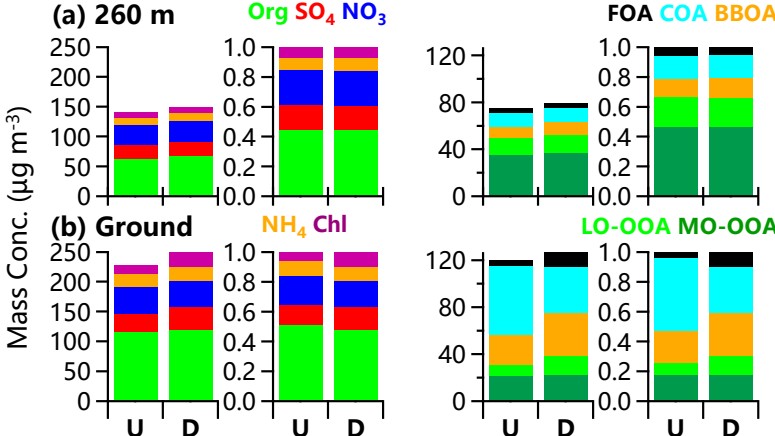

**Figure 9:** Average chemical composition of NR-PM$_1$ and OA at ground level and 260 m for vertical profile 11 (V11). Right panels are the mass fractions of NR-PM$_1$ and OA species. "U" and "D" represent "up" and "down" experiment, respectively.


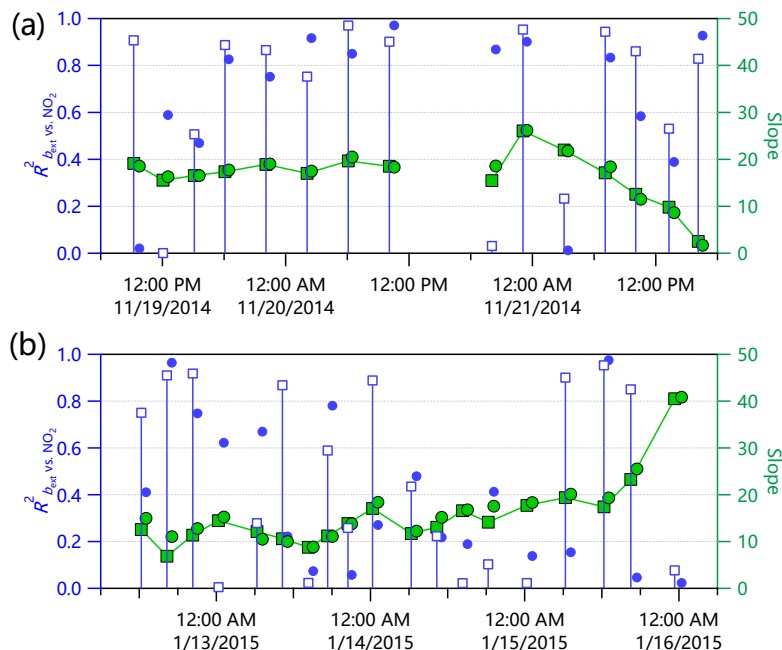

**Figure 10. Time series of correlation coefficients ($R^2$) and slopes between $b_{ext}$ and NO$_2$ for each vertical profile in (a) November and (b) January. The circles refer to the down profiles while other are the up profiles.**




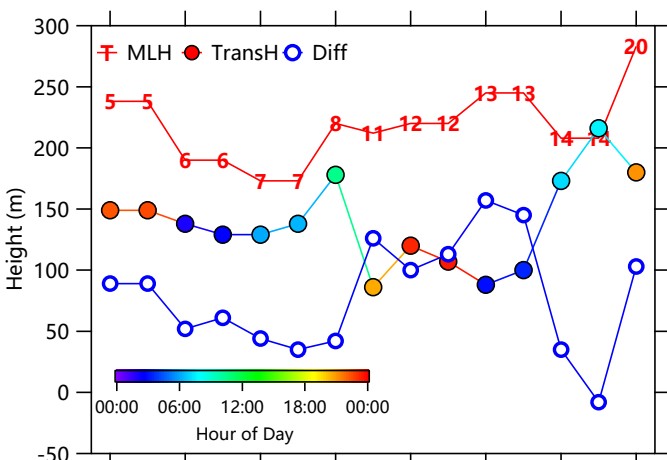

**Figure 11:** A comparison of mixing layer height (MLH) with that estimated from the transition heights of $b_{ext}$. The numbers of vertical profiles and the differences are also shown in the plot.