# Peer review of "Vertically-resolved characteristics of air pollution during two severe winter haze episodes in urban Beijing, China"

_Atmospheric Chemistry and Physics, 2017_

## Referee Comment (RC1) · Anonymous Referee #2 · 29 Nov 2017

General comments:

This manuscript reports results obtained during two haze episodes at Beijing in November 2014 and January 2015. The authors deployed a set of instruments at ground level and on the top of the Beijing Meteorological Tower (260 m) to measure the vertical profile of a few selected parameters (light extinction coefficient, $NO_2$, black carbon, non-refractory PM1, meteorological data).

This manuscript is very well written, and is totally relevant for the readers of Atmospheric Chemistry and Physics. I think that the manuscript in its current version is already in a very good shape. However, I have a few minor comments that the authors

may consider before final publication.

Specific comments:

1) Given that the authors are comparing concentrations between ground level and 260 m altitude, I'm wondering whether they need to convert the concentrations in standard temperature and pressure (STP) before doing the comparisons. I know that when people compare aircraft measurements to ground level data, the conversion to STP volumes is very important. Here, between ground level and 260 m, I'm not sure whether the difference of pressure requires this conversion. Among all the parameters that were measured at both altitudes, pressure is the only one for which the vertical profile is not given in this manuscript. I would suggest that the authors include it in the supplementary material (for instance in Figures S1 and S2), and check whether it's worth adjusting the concentrations to standard conditions.

2) Given that the chemical composition of non-refractory PM1 was measured with an HR-ToF-AMS at ground site and an ACSM on the top of the tower, I would suggest that the authors say a few words on their uncertainties. They can refer to the work of Crenn et al. (2015), who compared a set of 13 ACSMs with an HR-ToF-AMS. Then, the authors can check whether the differences in terms of concentrations and compositions observed during their study are significant, or whether they are within the uncertainties of the instruments.

3) Still concerning these two instruments, I'm curious to know how the authors performed the PMF analysis for the ACSM. Did they use the results obtained with the HR-ToF-AMS to choose the final PMF result for the ACSM (number of factors and fPeak)? I think that the authors need to add some evaluation plots for the 4-, 5-, and 6-factor solutions in the supplementary material, in order to justify the choice of the 5-factor solution for the two instruments. Here also, the authors can refer to the same inter-comparison between the HR-ToF-AMS and ACSMs (Fröhlich et al., 2015). In that work, the authors had noticed that some PMF factors can be quite difficult to separate

in some ACSM datasets (especially the separation of COA from HOA).

4) Among the five PMF factors identified with the HR-ToF-AMS and ACSM, three factors correspond to primary particles directly emitted at ground level (FOA, COA, BBOA). I'm wondering whether the authors can do a comparison between their mass spectra (Figure S3), and check whether some specific signals changed significantly between ground level and 260 m (following photo-oxidation, for instance). I guess this comparison should be quite difficult, given that the instrument on the top of the tower was the ACSM (mass spectra in unit mass resolution).

5) When I take a look at the vertical profiles of temperature during the 36 periods (Figures S1 and S2), I notice a strong decrease of the temperature at high altitude for at least 23 of these periods. This kind of vertical profile can have an incidence on the gas-particle partitioning of a few semi-volatile species (I'm especially thinking about nitrate), which can condense more at high altitude. The authors can include a discussion on this in the manuscript, for instance on page 11, lines 5-6, where the authors mention a higher contribution of nitrate at 260 m.

Technical comments:

6) Page 10, line 28: The SSA values are given in a certain range (i.e. min-max) for Fall 2014 and avg ± std dev for Winter 2015. Please choose one of the formats and use the same for the two periods, just to be consistent.

7) Caption of Figure 5: "260 m (top panel). The".

References:

Crenn, V., Sciare, J., Croteau, P. L., Verlhac, S., Fröhlich, R., Belis, C. A., Aas, W., Äijälä, M., Alastuey, A., Artiñano, B., Baisnée, D., Bonnaire, N., Bressi, M., Canagaratna, M., Canonaco, F., Carbone, C., Cavalli, F., Coz, E., Cubison, M. J., Esser-Gietl, J. K., Green, D. C., Gros, V., Heikkinen, L., Herrmann, H., Lunder, C., Minguillón, M. C., Močnik, G., O'Dowd, C. D., Ovadnevaite, J., Petit, J. E., Petralia, E., Poulain, L., Priest-

man, M., Riffault, V., Ripoll, A., Sarda-Estève, R., Slowik, J. G., Setyan, A., Wiedensohler, A., Baltensperger, U., Prévôt, A. S. H., Jayne, J. T., and Favez, O.: ACTRIS ACSM intercomparison – Part 1: Reproducibility of concentration and fragment results from 13 individual Quadrupole Aerosol Chemical Speciation Monitors (Q-ACSM) and consistency with co-located instruments, Atmos. Meas. Tech., 8, 5063-5087, 10.5194/amt-8-5063-2015, 2015.

Fröhlich, R., Crenn, V., Setyan, A., Belis, C. A., Canonaco, F., Favez, O., Riffault, V., Slowik, J. G., Aas, W., Aijälä, M., Alastuey, A., Artiñano, B., Bonnaire, N., Bozzetti, C., Bressi, M., Carbone, C., Coz, E., Croteau, P. L., Cubison, M. J., Esser-Gietl, J. K., Green, D. C., Gros, V., Heikkinen, L., Herrmann, H., Jayne, J. T., Lunder, C. R., Minguillón, M. C., Močnik, G., O'Dowd, C. D., Ovadnevaite, J., Petralia, E., Poulain, L., Priestman, M., Ripoll, A., Sarda-Estève, R., Wiedensohler, A., Baltensperger, U., Sciare, J., and Prévôt, A. S. H.: ACTRIS ACSM intercomparison – Part 2: Intercomparison of ME-2 organic source apportionment results from 15 individual, co-located aerosol mass spectrometers, Atmos. Meas. Tech., 8, 2555-2576, 10.5194/amt-8-2555-2015, 2015.

---

## Referee Comment (RC2) · Anonymous Referee #1 · 5 Dec 2017

The paper presents real-time continuous vertical measurements of particle extinction, gaseous NO2, and black carbon (BC) from ground level to 260 m during two severe winter haze episodes at an urban site in Beijing, China. This study is very interesting and important in helping understand the formation mechanisms and evolution of severe haze episodes in China. I have a few minor issues to be considered before accepting this paper for publication. 1, P1 L14-16, it states "there were four types of vertical profiles with different occurrence rate", but 37%+5%+29%+16%+14%=101% not 100%? 2, The paper states "The travel height is 260m and the container travel at a constant speed of 8 m/s", so it takes 32.5 min to travel up and another 32.5 min to travel back down. But the time in Table S1 shows a very inconsistent travel time, some

have travel time of 28.5 min, some have 26 min. Why is that? 3, Because the measurements at different heights were not measured simultaneously (it had a ∼30 min lag), the sources and composition of aerosol may change in ∼30 min especially for local source, leading to biased vertical difference. How to address this? 4, P4, L12, it states "fossil OA (FOA) is predominantly from coal combustion emissions", why is this? Is it not possible to resolve traffic HOA in Beijing using HR-AMS and ACSM-PMF? 5, The ACTRIS ACSM intercomparison study (Crenn et al., 2015) shows that ACSM and HR-AMS measurement would vary (e.g., 36% for ammonium) to some degree even when measuring from the same inlet. Here in this study, the vertical difference between ACSM and HR-AMS would be less meaningful if they would already vary a lot. Do you have inter comparison study between ACSM and HR-AMS measuring from the same inlet?

---

## Short Comment (SC1) · 14 Dec 2017

I agree with the two anonymous reviewers that this is an interesting and well written paper. This concerns not only the "chemical" aspects mentioned but also the determination of the mixing layer height (MLH): it is estimated from extinction coefficient profiles (aethalometer onboard of a vertically moving container) and attenuated backscatter profiles (Vaisala CL51 ceilometer). Though both parameters are different they offer an excellent opportunity for intercomparisons, as both are related to aerosol optical properties. Consequently, the authors briefly cover this topic and conclude that the MLH tends to be overestimated when using the ceilometer.

[Figure]

I suggest to discuss this interesting application of the data in more depth:

- From the reference Tang et al. (2016) it can be inferred that the authors use BL-VIEW. It has been shown by Geiß et al. (2017, Atmos. Meas. Tech.), that depending on the different options the retrieved MLH can be different. Thus, it would be interesting to include a few additional information on how the authors determine the MLH. By the way: Geiß et al. also found that the different versions of BL-VIEW tend to (slightly) overestimate the MLH.

  The quantitative criteria underlying the determination of the "transition height" from $b_{ext}$ should be outlined as well (or be stated, that it is from visual inspection).

- A figure showing all coincident MLH-retrievals would be interesting. Fig. 11 – only shown in the conclusions – seems to provide this. I assume that "5", "6" etc. along with the MLH-curve correspond to Table S1? As the number of cases is relatively low (because quite often the MLH is larger than 260 m) this is not obvious and thus should be clearly emphasized. Maybe, the figure should be moved to the results-section. Fig. 7e is less suitable to demonstrate the differences as it does not cover the full set of measurements, and the reader might be confused from the two different vertical scales.

- It is known that an overlap correction function is applied to ceilometer measurements, well below 260 m in case of the CL51. Consequently comparisons with the independent extinction coefficient measurements could be a promising approach to check the plausibility of this correction, whenever adequate atmospheric conditions occur (e.g., no rapid changes of the aerosol distribution). Though a strict validation might be difficult and beyond the scope of this paper, it could be briefly discussed in section 4 whether or not this would be a possible extension of this study.

- A "definition" of "severe haze episodes" in terms of aerosol optical depth would
be interesting: In case of very large AOD the ceilometer might not fully penetrate the mixing layer. From the extinction coefficient profiles (aethalometer) it seems that the AOD is however clearly below 1 (and thus not critical). Do conditions occur in Beijing when this is not true?

Technical comments:

- Page 5, line 32: include brackets around "2 $\times$ mean".

- Page 6, line 20: when referring to V3 and V24 etc. it would be useful to mention Table S1 again, where the nomenclature is (more or less) explained.

- Define the light green and the dark green curve in Fig. 7e.

---

## Author Comment (AC1) · 29 Dec 2017

We thank the three reviewers for their constructive comments, which helped improve the manuscript substantially. We have revised the manuscript according to the reviewers' comments. Listed below is our point-to-point response in blue to each comment that was offered by the reviewers.

**Response to Reviewer #1**

Comments:

The paper presents real-time continuous vertical measurements of particle extinction, gaseous NO2, and black carbon (BC) from ground level to 260 m during two severe winter haze episodes at an urban site in Beijing, China. This study is very interesting and important in helping understand the formation mechanisms and evolution of severe haze episodes in China. I have a few minor issues to be considered before accepting this paper for publication.

1, P1 L14-16, it states "there were four types of vertical profiles with different occurrence rate", but 37%+5%+29%+16%+14%=101% not 100%?

Thank the reviewer's carefulness. 5% here refers to the vertical differences rather than occurrence rate. The four types of vertical profiles account for 37%, 29%, 16%, and 15%, respectively of the total number of vertical profiles in this study, and they together account for 96% with the rest of 4% being other types.

2, The paper states "The travel height is 260m and the container travel at a constant speed of 8 m/s", so it takes 32.5 min to travel up and another 32.5 min to travel back down. But the time in Table S1 shows a very inconsistent travel time, some have travel time of 28.5 min, some have 26 min. Why is that?

Thank the reviewer's comments. In fact, the speed is approximately 9 m min$^{-1}$. The time was not very consistent for each vertical profile because (1) the travel of the

container is controlled by a mechanical gear that failed several times in the middle of the experiments, (2) the container was left at different heights (sometimes 10 m high, and sometimes at ground level) after the "down" experiment. Following the reviewer's comments, we revised the sentence in the new version of the manuscript. It now reads:

"using a container that can travel on the BMT at a relatively constant speed of approximately 9 m min$^{-1}$"

3, Because the measurements at different heights were not measured simultaneously (it had a 30 min lag), the sources and composition of aerosol may change in 30 min especially for local source, leading to biased vertical difference. How to address this?

Thank the reviewer for pointing this out. In this study, we analyzed each vertical profile, in particular those with significant changes as a function of height. To investigate the reasons for the vertical changes, we then analyzed the vertical profiles of meteorological parameters, and also compared the time series of aerosol species measured at ground level and 260 m. We found that the changes in vertical profiles were dominantly associated with meteorological conditions. We agree with the reviewer that the time-lag of local source emissions could affect the vertical profiles. According to previous studies (Sun et al., 2013;Sun et al., 2015), the dominant local source with dramatic changes in a short time is cooking emission. For example, during the period of V11, we observed a large increase in COA in 1 hour. However, such an increase appears not affect the vertical profiles of particle extinction and BC. The reason is that COA has minor impacts on light extinction, and cooking emission is not an important source of BC (Han et al., 2015;Wang et al., 2015;He et al., 2004). Because we didn't measure the vertical profiles of aerosol composition, the time-lag effect could not affect the vertical profiles of light extinction, BC, and $NO_2$ substantially.

We did observe a very few vertical profiles that were affected by the time-lag effect.

For example, a sudden increase in $b_{ext}$ was observed in V19 when there was no $T$ inversion, RH was constant across different heights, and WS was higher at higher altitudes. We found that such vertical profiles were mainly caused by the rapid increase in PM pollution within a short time (the concentration of NR-PM$_1$ increased from 30 to 127 µg m$^{-3}$ within approximately 20 min from 16:30 to 16:50 on January 12, 2015, Figure 2j). The V19 experiment started at 16:19 when the NR-PM$_1$ concentration was still low and the increase in the middle (~170 m) coincided with the rapid increase of PM pollution, leading to a significant change in $b_{ext}$.

Considering the reasons above, the time-lag effect would not affect our conclusions, and the vertical profiles are expected to representative.

4, P4, L12, it states "fossil OA (FOA) is predominantly from coal combustion emissions", why is this? s it not possible to resolve traffic HOA in Beijing using HR-AMS and ACSM-PMF?

Yes, it is very challenging to separate the traffic-related HOA from coal combustion OA (CCOA) through PMF analysis of unit mass resolution mass spectra of either HR-ToF-AMS or ACSM. The reasons include: (1) very similar spectral patterns between HOA and CCOA at $m/z$ < 120; (2) very similar temporal variations and diurnal cycles; (3) limited sensitivity of the ACSM, particularly for $m/z$'s > 50 with large uncertainties in ion transmission efficiencies. Sun et al. (2016) was able to separate HOA from CCOA by using PMF analysis of high resolution mass spectra and the UMR spectra to $m/z$ = 350, while they were not separated in this study even extending the solution to 6 or 7 factors. Therefore, the two sources are combined into one factor, i.e., fossil fuel related OA (FFOA). Coal combustion OA was the most important primary OA in winter in Beijing, which is much higher than that of HOA (Hu et al., 2016;Sun et al., 2016;Sun et al., 2013). One of the reasons is due to the largely enhanced coal combustion emissions for residential heating while the diesel trucks and heavy-duty vehicles are only allowed inside the Beijing city between 23:00 – 6:00.

5, The ACTRIS ACSM intercomparison study (Crenn et al., 2015) shows that ACSM and HR-AMS measurement would vary (e.g., 36% for ammonium) to some degree even when measuring from the same inlet. Here in this study, the vertical difference between ACSM and HR-AMS would be less meaningful if they would already vary a lot. Do you have inter comparison study between ACSM and HR-AMS measuring from the same inlet?

Good point. We did a two-week inter-comparison between HR-ToF-AMS and ACSM measurements before this study. Although both ACSM and HR-ToF-AMS were calibrated, the ACSM measurements were further corrected using the regression slopes against HR-ToF-AMS measurements from the inter-comparisons to reduce the uncertainties in vertical comparisons.

Following the reviewer's suggestions, we added more details on the inter-comparisons in the revised manuscript. It now reads:

"Considering that the ACSM measurements can have uncertainties of 9 – 36% for different NR-PM$_1$ species (Crenn et al., 2015), we performed a two-week inter-comparison between ACSM and HR-ToF-AMS measurements at ground site. All submicron aerosol species measured by the ACSM were highly correlated with those measured by the HR-ToF-AMS ($R^2 > 0.97$), and the regression slopes of ACSM against HR-AMS varied from 0.61 to 1.24 for different aerosol species (Sun et al., 2015). To reduce the uncertainties in vertical comparisons, the ACSM measurements were further corrected using the regression slopes determined from the inter-comparisons."

**Response to Reviewer #2**

Comments:

General comments:

This manuscript reports results obtained during two haze episodes at Beijing in November 2014 and January 2015. The authors deployed a set of instruments at ground level and on the top of the Beijing Meteorological Tower (260 m) to measure the vertical profile of a few selected parameters (light extinction coefficient, NO2, black carbon, non-refractory PM1, meteorological data). This manuscript is very well written, and is totally relevant for the readers of Atmospheric Chemistry and Physics. I think that the manuscript in its current version is already in a very good shape. However, I have a few minor comments that the authors may consider before final publication.

Specific comments:

1) Given that the authors are comparing concentrations between ground level and 260 m altitude, I'm wondering whether they need to convert the concentrations in standard temperature and pressure (STP) before doing the comparisons. I know that when people compare aircraft measurements to ground level data, the conversion to STP volumes is very important. Here, between ground level and 260 m, I'm not sure whether the difference of pressure requires this conversion. Among all the parameters that were measured at both altitudes, pressure is the only one for which the vertical profile is not given in this manuscript. I would suggest that the authors include it in the supplementary material (for instance in Figures S1 and S2), and check whether it's worth adjusting the concentrations to standard conditions.

[Figure]

Figure R1 (a) Pressure ($P$), (b) $T$ at 16 m and 280 m, and (c) the ratios of $P/T$ at 280 m and 16 m.

We thank the reviewer for pointing this out. The pressure was measured at 16 m and 280 m in this study, which was used to evaluate the impacts of pressure on vertical differences. We calculated the ratios of pressure/temperature at the two heights (according to ideal gas law). As shown in Figure R1, the vertical difference caused by the pressure and temperature were both less than 4% during the two severe haze episodes, and the average differences are 2.8% and 2.4% in November and January. Such differences are much smaller than the measurement uncertainties of the ACSM and HR-ToF-AMS, we therefore did not convert the measurements at 260 m to those under standard temperature and pressure conditions.

2) Given that the chemical composition of non-refractory PM1 was measured with an HR-ToF-AMS at ground site and an ACSM on the top of the tower, I would suggest that the authors say a few words on their uncertainties. They can refer to the work of Crenn et al. (2015), who compared a set of 13 ACSMs with an HR-ToF-AMS. Then, the authors can check whether the differences in terms of concentrations and

compositions observed during their study are significant, or whether they are within the uncertainties of the instruments.

Thank the reviewer's comments. We did a two-week inter-comparison between HR-ToF-AMS and ACSM measurements before this study. Although both ACSM and HR-ToF-AMS were calibrated, the ACSM measurements were further corrected using the regression slopes against HR-ToF-AMS measurements from the inter-comparisons to reduce the uncertainties in vertical comparisons.

Following the reviewer's suggestions, we added more details on the inter-comparisons in the revised manuscript. It now reads:

"Considering that the ACSM measurements can have uncertainties of 9 – 36% for different NR-PM$_1$ species (Crenn et al., 2015), we performed a two-week inter-comparison between ACSM and HR-ToF-AMS measurements at ground site. All submicron aerosol species measured by the ACSM were highly correlated with those measured by the HR-ToF-AMS ($R^2 > 0.97$), and the regression slopes of ACSM against HR-AMS varied from 0.61 to 1.24 for different aerosol species (Sun et al., 2015). To reduce the uncertainties in vertical comparisons, the ACSM measurements were further corrected using the regression slopes determined from the inter-comparisons."

3) Still concerning these two instruments, I'm curious to know how the authors performed the PMF analysis for the ACSM. Did they use the results obtained with the HR-ToF-AMS to choose the final PMF result for the ACSM (number of factors and fPeak)? I think that the authors need to add some evaluation plots for the 4-, 5-, and 6-factor solutions in the supplementary material, in order to justify the choice of the 5-factor solution for the two instruments. Here also, the authors can refer to the same inter-comparison between the HR-ToF-AMS and ACSMs (Fröhlich et al., 2015). In that work, the authors had noticed that some PMF factors can be quite difficult to separate in some ACSM datasets (especially the separation of COA from HOA).

We thank the reviewer's comments. We expanded the details on PMF and ME2 analysis substantially in the revised manuscript. The detailed evaluation of PMF results and the a-value based ME-2 solutions were given in Zhou et al. (2017).

"Positive matrix factorization (PMF) (Paatero and Tapper, 1994) was first performed to the unit mass resolution spectra of OA at ground level that were measured with HR-ToF-AMS during the same period as that of ACSM, and five factors including three primary OA (POA) factors, i.e., fossil fuel related OA (FFOA) predominantly from coal combustion emissions, cooking OA (COA), and biomass burning OA (BBOA), and two secondary OA (SOA) factors, i.e., less oxidized oxygenated OA (LO-OOA) and more oxidized OOA (MO-OOA) were identified. To better compare the OA factors between ground level and 260 m, the multi-linear engine 2 (ME-2) (Canonaco et al., 2013;Crippa et al., 2014) using the mass spectral profiles of three POA factors at ground level as constrains was performed to the ACSM OA spectra. In addition to the three POA factors, a LO-OOA and a MO-OOA were also resolved. It should be noted that such an approach could introduce some uncertainties for OA source apportionment at 260 m because POA factors are not exactly the same between ground level and 260 m. We also performed PMF analysis on ACSM OA spectra, and found that the BBOA factor cannot be resolved although biomass burning is a common source in winter. The detailed evaluation of PMF results and the a-value based ME-2 solutions were given in Zhou et al. (2017)."

4) Among the five PMF factors identified with the HR-ToF-AMS and ACSM, three factors correspond to primary particles directly emitted at ground level (FOA, COA, BBOA). I'm wondering whether the authors can do a comparison between their mass spectra (Figure S3), and check whether some specific signals changed significantly between ground level and 260 m (following photo-oxidation, for instance). I guess this comparison should be quite difficult, given that the instrument on the top of the tower was the ACSM (mass spectra in unit mass resolution).

Thank the reviewer's comments. The three POA factors at 260 m were determined

using a-value based ME-2 analysis. The mass spectra profiles of three POA factors resolved at ground site were used as constrains. In addition, PMF analysis was also performed to ACSM OA spectra, and only two primary OA factors were identified. The comparisons of mass spectral profiles for different a-values and also PMF results are shown in Figure R2. More detailed descriptions of the results are given in Zhou et al. (2017).

[Figure]

[Figure]

Figure R2: Mass spectra (left panel) and time series (right panel) of three POA factors resolved at 260 m by ACSM using multi-linear engine 2 (ME-2): (a) fossil fuel related OA (FFOA), (b) cooking OA (COA), and (c) biomass-burning OA (BBOA). The 4-factor solution of PMF results is also shown.

5) When I take a look at the vertical profiles of temperature during the 36 periods (Figures S1 and S2), I notice a strong decrease of the temperature at high altitude for at least 23 of these periods. This kind of vertical profile can have an incidence on the gas-particle partitioning of a few semi-volatile species (I'm especially thinking about

nitrate), which can condense more at high altitude. The authors can include discussion on this in the manuscript, for instance on page 11, lines 5-6, where the authors mention a higher contribution of nitrate at 260 m.

Thank the reviewer's careful review. We agree with the reviewer that gas-particle partitioning can have a significant impact on the formation of nitrate. We note that the highest temperature during the two severe haze episodes was approximately $15^{o}C$ and $8^{o}C$, respectively. The evaporative loss of ammonium nitrate under such low temperature conditions could not be important. Therefore, gas-particle partitioning formation of nitrate is not expected to be as important as that observed in summer. Comparatively, the vertical differences in $O_3$, $NO_2$, and solar radiation could be more important for the nitrate differences via heterogeneous reactions of $N_2O_5$ at nighttime and photochemical production during daytime. Unfortunately, we didn't have such measurements to evaluate such impacts. As the reviewer mentioned, the higher nitrate contribution at 260 m (page 11, lines 5-6) was mainly caused by the lower contribution of organics. In fact, the absolute nitrate concentration at 260 m was lower than that at ground site. The large increase in OA from local sources at ground site led to a decrease of relative contribution of nitrate in NR-PM$_1$. Similarly, the sulfate contribution at ground site is also lower than that at 260 m. Therefore, it is very challenging to quantify the impact of gas-particle partitioning on the vertical differences of nitrate in this study. Such impacts should be investigated with more comprehensive measurements in the future studies.

Technical comments:

6) Page 10, line 28: The SSA values are given in a certain range (i.e. min-max) for Fall 2014 and avg    std dev for Winter 2015. Please choose one of the formats and use the same for the two periods, just to be consistent.

It was revised.

7) Caption of Figure 5: "260 m (top panel). The".

Corrected.

**Response to Prof. M. Wiegner**

I agree with the two anonymous reviewers that this is an interesting and well written paper. This concerns not only the "chemical" aspects mentioned but also the determination of the mixing layer height (MLH): it is estimated from extinction coefficient profiles (aethalometer onboard of a vertically moving container) and attenuated backscatter profiles (Vaisala CL51 ceilometer). Though both parameters are different they offer an excellent opportunity for intercomparisons, as both are related to aerosol optical properties. Consequently, the authors briefly cover this topic and conclude that the MLH tends to be overestimated when using the ceilometer. I suggest to discuss this interesting application of the data in more depth:

Thank Prof. M. Wiegner for your comments.

• From the reference Tang et al. (2016) it can be inferred that the authors use BL-VIEW. It has been shown by Geiß et al. (2017, Atmos. Meas. Tech.), that depending on the different options the retrieved MLH can be different. Thus, it would be interesting to include a few additional information on how the authors determine the MLH. By the way: Geiß et al. also found that the different versions of BL-VIEW tend to (slightly) overestimate the MLH. The quantitative criteria underlying the determination of the "transition height" from $b_{ext}$ should be outlined as well (or be stated, that it is from visual inspection).

We thank the reviewer's careful review. Following the reviewer's suggestions, we added more information about the retrieval method of CL51 data for MLH in our study.

It now reads:

"The Vaisala software product BL-VIEW (version 2.0) was used to identify the MLH with the gradient method. The temporal and vertical attenuated backscatter coefficients were first smoothly averaged to avoid the effect of noise and interference

from the aerosol layering structure, and the maximum negative gradient value ($-\mathrm{d}\beta/\mathrm{d}x$) was then determined as the top of the mixing layer (Münkel et al., 2007;Geiß et al., 2017;Zhu et al., 2016)."

The transition height was determined mainly from the visual perspective, which was added in the revised manuscript. An uncertainty of ~5 m is expected.

• A figure showing all coincident MLH-retrievals would be interesting. Fig. 11 – only shown in the conclusions – seems to provide this. I assume that "5", "6" etc. along with the MLH-curve correspond to Table S1? As the number of cases is relatively low (because quite often the MLH is larger than 260 m) this is not obvious and thus should be clearly emphasized. Maybe, the figure should be moved to the results-section. Fig. 7e is less suitable to demonstrate the differences as it does not cover the full set of measurements, and the reader might be confused from the two different vertical scales.

Thank the reviewer's comments. The numbers in Fig. 11 corresponded to those in Table S1 and Figure 1. We agree with the reviewer that the number of cases is relatively low because only vertical profiles with significant changes can be used to estimate the transition heights. Following the reviewer's suggestions, we clearly stated this point in the revised manuscript.

"Because of the relatively low number of cases and the limited height of the meteorological tower, future vertical measurements of particle extinction to a high altitude, e.g., using tethered balloon are needed to further validate the retrieval of MLH from CL51 measurements."

Fig. 7e is mainly used to demonstrate the evolution of vertical profiles during the severe haze episode. For clarity, we claimed "right axis" in the figure caption. Also, a clearer description "Note that only vertical profiles with significant changes were used for estimation of transition heights." was added in Figure 11.

• It is known that an overlap correction function is applied to ceilometer measurements, well below 260 m in case of the CL51. Consequently, comparisons with the independent extinction coefficient measurements could be a promising approach to check the plausibility of this correction, whenever adequate atmospheric conditions occur (e.g., no rapid changes of the aerosol distribution). Though a strict validation might be difficult and beyond the scope of this paper, it could be briefly discussed in section 4 whether or not this would be a possible extension of this study.

We totally agree with the reviewer. "future vertical measurements of particle extinction to a high altitude, e.g., using tethered balloon are needed to further validate the retrieval of MLH from CL51 measurements" was added in the revised text.

• A "definition" of "severe haze episodes" in terms of aerosol optical depth would be interesting: In case of very large AOD the ceilometer might not fully penetrate the mixing layer. From the extinction coefficient profiles (aethalometer) it seems that the AOD is however clearly below 1 (and thus not critical). Do conditions occur in Beijing when this is not true?

Yes, it is interesting to check the relationship between air pollution level and AOD. Indeed, several previous studies found the positive correlations between AOD and $PM_{2.5}$, even at high $PM_{2.5}$ levels (> 200 µg m$^{-3}$), and the slopes are strongly regional dependent. Therefore, AOD could be used to judge the severe haze episodes. Concerning the penetration of mixing layer during severe haze episodes, the CL51 appears to work relatively well as indicated by Figure R3 when the NR-$PM_1$ was the highest in November. However, an accurate evaluation of such impacts needs vertical measurements to a high altitude, for example, > 1 km.

[Figure]

Figure R3. Vertical distribution of attenuated backscatter coefficient on 20 November, 2014.

Technical comments:

• Page 5, line 32: include brackets around "2 mean".

Corrected.

• Page 6, line 20: when referring to V3 and V24 etc. it would be useful to mention Table S1 again, where the nomenclature is (more or less) explained.

Table S1 was added.

• Define the light green and the dark green curve in Fig. 7e.

The two green lines refer to the time series of NR-PM$_1$ at ground level and 260 m, which are described in the figure legend (above Fig. 7e).

References

Canonaco, F., Crippa, M., Slowik, J. G., Baltensperger, U., and Prévôt, A. S. H.: SoFi, an IGOR-based interface for the efficient use of the generalized multilinear engine (ME-2) for the source apportionment: ME-2 application to aerosol mass spectrometer data, Atmos. Meas. Tech., 6, 3649-3661, 2013.

Crenn, V., Sciare, J., Croteau, P. L., Verlhac, S., Fröhlich, R., Belis, C. A., Aas, W., Äijälä, M., Alastuey, A., Artiñano, B., Baisnée, D., Bonnaire, N., Bressi, M., Canagaratna, M., Canonaco, F., Carbone, C., Cavalli, F., Coz, E., Cubison, M. J., Esser-Gietl, J. K., Green, D. C., Gros, V., Heikkinen, L., Herrmann, H., Lunder, C., Minguillón, M. C., Močnik, G., O'Dowd, C. D., Ovadnevaite, J., Petit, J. E., Petralia, E., Poulain, L., Priestman, M., Riffault, V., Ripoll, A., Sarda-Estève, R., Slowik, J. G., Setyan, A., Wiedensohler, A., Baltensperger, U., Prévôt, A. S. H., Jayne, J. T., and Favez, O.: ACTRIS ACSM intercomparison – Part 1: Reproducibility of concentration and fragment results from 13 individual Quadrupole Aerosol Chemical Speciation Monitors (Q-ACSM) and consistency with co-located instruments, Atmos. Meas. Tech., 8, 5063-5087, 10.5194/amt-8-5063-2015, 2015.

Crippa, M., Canonaco, F., Lanz, V. A., Äijälä, M., Allan, J. D., Carbone, S., Capes, G., Ceburnis, D., Dall'Osto, M., and Day, D. A.: Organic aerosol components derived from 25 AMS data sets across Europe using a consistent ME-2 based source apportionment approach, Atmos. Chem. Phys., 14, 6159-6176, 2014.

Geiß, A., Wiegner, M., Bonn, B., Schäfer, K., Forkel, R., von Schneidemesser, E., Münkel, C., Chan, K. L., and Nothard, R.: Mixing layer height as an indicator for urban air quality?, Atmos. Meas. Tech., 10, 2969-2988, 10.5194/amt-10-2969-2017, 2017.

Han, T., Xu, W., Chen, C., Liu, X., Wang, Q., Li, J., Zhao, X., Du, W., Wang, Z., and Sun, Y.: Chemical apportionment of aerosol optical properties during the Asia-Pacific Economic Cooperation (APEC) summit in Beijing, China, J. Geophys. Res., 120, 15, 12281-12295, doi:10.1002/2015JD023918., 2015.

He, L. Y., Hu, M., Huang, X. F., Yu, B. D., Zhang, Y. H., and Liu, D. Q.: Measurement of emissions of fine particulate organic matter from Chinese cooking, Atmos. Environ., 38, 6557-6564, 2004.

Hu, W., Hu, M., Hu, W., Jimenez, J. L., Yuan, B., Chen, W., Wang, M., Wu, Y., Chen, C., and Wang, Z.: Chemical composition, sources, and aging process of submicron aerosols in Beijing: Contrast between summer and winter, J. Geophys. Res., 121, 1955-1977, 2016.

Münkel, C., Eresmaa, N., Räsänen, J., and Karppinen, A.: Retrieval of mixing height and dust concentration with lidar ceilometer, Bound-Lay. Meteorol., 124, 117-128, 2007.

Paatero, P., and Tapper, U.: Positive matrix factorization - A nonnegative factor model with optimal utilization of error estimates of data values Environmetrics, 5, 111-126, 10.1002/env.3170050203, 1994.

Sun, Y., Wang, Z., Fu, P., Yang, T., Jiang, Q., Dong, H., Li, J., and Jia, J.: Aerosol

composition, sources and processes during wintertime in Beijing, China, Atmos. Chem. Phys., 13, 4577-4592, 2013.

Sun, Y., Wei, D., Wang, Q., Zhang, Q., Chen, C., Chen, Y., Chen, Z., Fu, P., Wang, Z., and Gao, Z.: Real-Time Characterization of Aerosol Particle Composition above the Urban Canopy in Beijing: Insights into the Interactions between the Atmospheric Boundary Layer and Aerosol Chemistry, Environ. Sci. Technol., 49, 11340-11347, 2015.

Sun, Y., Du, W., Fu, P., Wang, Q., Li, J., Ge, X., Zhang, Q., Zhu, C., Ren, L., Xu, W., Zhao, J., Han, T., Worsnop, D. R., and Wang, Z.: Primary and secondary aerosols in Beijing in winter: sources, variations and processes, Atmos. Chem. Phys., 16, 8309-8329, 10.5194/acp-16-8309-2016, 2016.

Wang, Q., Sun, Y., Jiang, Q., Du, W., Sun, C., Fu, P., and Wang, Z.: Chemical composition of aerosol particles and light extinction apportionment before and during heating season in Beijing, China, J. Geophys. Res., 120, 12708-12722, 10.1002/2015JD023871, 2015.

Zhou, W., Wang, Q., Zhao, X., Xu, W., Chen, C., Du, W., Zhao, J., Canonaco, F., Prévôt, A. S. H., Fu, P., Wang, Z., Worsnop, D. R., and Sun, Y.: Characterization and source apportionment of organic aerosol at 260 m on a meteorological tower in Beijing, China, Atmos. Chem. Phys. Discuss., 2017, 1-34, 10.5194/acp-2017-1039, 2017.

Zhu, X., Tang, G., Hu, B., Wang, L., Xin, J., Zhang, J., Liu, Z., Münkel, C., and Wang, Y.: Regional pollution and its formation mechanism over North China Plain: A case study with ceilometer observations and model simulations, J. Geophys. Res., 121, 14574-14588, 2016.